# Investigation of a Liquid-Phase Electrode for Micro-Electro-Discharge Machining

**DOI:** 10.3390/mi11100935

**Published:** 2020-10-14

**Authors:** Ruining Huang, Ying Yi, Erlei Zhu, Xiaogang Xiong

**Affiliations:** 1School of Mechanical Engineering and Automation, Harbin Institute of Technology, Shenzhen 518000, China; zel15950568268@163.com (E.Z.); xiongxg@hit.edu.cn (X.X.); 2College of Science and Engineering, Hamad Bin Khalifa University, Education City 34110, Qatar; yyi@hbku.edu.qa

**Keywords:** Micro-electro-discharge machining (μEDM), liquid-metal electrode, Galinstan

## Abstract

Micro-electro-discharge machining (μEDM) plays a significant role in miniaturization. Complex electrode manufacturing and a high wear ratio are bottlenecks for μEDM and seriously restrict the manufacturing of microcomponents. To solve the electrode problems in traditional EDM, a µEDM method using liquid metal as the machining electrode was developed. Briefly, a liquid-metal tip was suspended at the end of a capillary nozzle and used as the discharge electrode for sparking the workpiece and removing workpiece material. During discharge, the liquid electrode was continuously supplied to the nozzle to eliminate the effects of liquid consumption on the erosion process. The forming process of a liquid-metal electrode tip and the influence of an applied external pressure and electric field on the electrode shape were theoretically analyzed. The effects of external pressure and electric field on the material removal rate (MRR), liquid-metal consumption rate (LMCR), and groove width were experimentally analyzed. Simulation results showed that the external pressure and electric field had a large influence on the electrode shape. Experimental results showed that the geometry and shape of the liquid-metal electrode could be controlled and constrained; furthermore, liquid consumption could be well compensated, which was very suitable for µEDM.

## 1. Introduction

Micro-electro-discharge machining (μEDM) is a powerful micromachining technique with various advantages resulting from it being a noncontact and thermal process; thus, µEDM is applicable to any electrically conductive material regardless of the mechanical properties of the material. The machining process utilizes the thermal erosion of the material caused by pulses of electrical discharge generated between a microscopic electrode and the workpiece in the presence of a dielectric fluid for the removal of the workpiece material. The µEDM technique is capable of producing microholes, microchannels, and real three-dimensional (3D) microstructures. These attractive features have been leveraged for producing micromechanical components, as well as for prototyping various micro-electro-mechanical systems (MEMS) and devices. Furthermore, µEDM has been widely used in the aerospace, die, mold, and biomedical industries for machining small cavities [1,2,3].

It is well known that electrode miniaturization is critical for μEDM because it determines the precision of the process to manufacture smaller and more precise parts [4]. Many methods have been proposed for fabricating microelectrodes. For example, Masuzawa et al. were the first to develop wire electrode discharge grinding (WEDG) technology to make a microelectrode with a diameter of Φ2.5 μm [5]. WEDG is one of the most widely used methods to fabricate microelectrodes, and it offers the benefit of sustaining good grinding accuracy by utilizing fresh wire during the whole fabrication process. According to this method, some evolutions occurred in different ways. Egashira et al. made use of WEDG and electrochemical machining to produce a microelectrode with a diameter of Φ0.3 μm [6]. Zhang et al. proposed a tangential feed WEDG (TF-WEDG) method to improve the microelectrode accuracy [7]. Lim et al. utilized a rotating disc instead of a moving wire to grind the microelectrode [8]. Other methods reported for μEDM electrode fabrication include the single-side block electrode discharge grinding method (BEDG) [9], bilateral BEDG method [10], scanning discharge method [11], single-pulse discharge method [12], self-drilled hole reverse-EDM method [13], electroforming method [14], Lithographie, Galvanoformung and Abformung (LIGA) process method [15], and electrostatic ejection method [16]. The goal of all these methods is the same: to improve the processing efficiency and quality, reduce the processing difficulty, and make smaller and better microelectrodes. However, these methods either are complex processes or require additional devices, leading to the process being cumbersome and inefficient. It is difficult to control the size and precision of the microelectrode. Although high-quality electrodes can sometimes be obtained, the consistency is poor, and it is difficult to obtain the same size electrode again.

In addition, the mechanism of μEDM is an electrothermal physical process that removes material by repeated spark discharges, whereby the workpiece is eroded at high temperature, and the electrode itself will also wear down. Therefore, the fundamental theory determines the unavoidability of wear with the electrode tool. Due to the area effect, with the decrease in electrode diameter, the relative electrode wear rate becomes more serious [17]. Some research work has focused on electrode wear issues during the μEDM process to obtain improved machining accuracy. For example, Bissacco et al. analyzed the electrode wear rate at different energy levels in detail [18]. Wang et al. carried out quantitative research on the electrode wear amount of positive and negative pulses [19]. Tsai et al. conducted a detailed investigation on the wear rate of electrodes of different materials [20]. All these investigations indicate that electrode wear will have a serious impact on the subsequent processing performance, resulting in a decrease in machining accuracy. To address a variety of problems associated with electrode wear, many explorations on preventing and compensating electrode wear have been performed. The measures reported for preventing μEDM wear include the ultrasonic-assisted debris removal method [2], coating electrode method [21], special material electrode method [22], and discharge in gas method [23]. The methods reported for μEDM electrode compensation include the electrode uniform wear compensation method [24], electrode fixed length linear compensation method [25], effective discharge pulse monitoring compensation method [26], and prediction electrode compensation method [27]. However, the above methods are only suitable for specific occasions and have certain limitations, which lead to imperfect final results and difficulty in the promotion of these methods in industrial fields. Moreover, it is difficult to measure the actual wear of microelectrodes online, and the consistency of the online electrode is difficult to guarantee. Therefore, electrode compensation has always been one of the difficult problems in μEDM, and the various electrode compensation methods have not completely solved the problem caused by electrode wear. Thus, it is urgent to find a new processing method to solve the issues of electrode wear.

To address the wear-related problems associated with μEDM, Huang et al. proposed a novel μEDM method that uses a liquid alloy as the machining electrode instead of traditional electrodes [17,28], in which the liquid metal consumed in the process can be compensated over time, and the capillary containing the liquid metal does not participate in the discharge; thus, its shape remains unchanged and solves the problem of electrode wear. Nevertheless, these references reported the experimental characterizations and demonstration of the process on the basis of a preliminary study and lacked an analysis of the liquid-metal electrode morphology, which will affect the machining accuracy. Therefore, this study specifically focuses on analyzing the influence of the external pressure and electric field on the shape of liquid electrodes, as well as the influence of different electrode shapes on machining characteristics. The characterization of the process with varying discharge parameters is discussed, along with the arbitrary patterning of silicon substrates using the developed method.

## 2. Method

The principle of conducting μEDM with a liquid-metal electrode is illustrated in Figure 1. A liquid metal is held through a metallic capillary nozzle coated with a dielectric film so that the liquid protrudes on the nozzle tip and forms a droplet. Under the action of an appropriate pressure and electric field, the droplet suspended at the tip will countervail part of the surface tension and become a conical tip that serves as a microdischarge electrode. A pulse generator is applied between the liquid metal and the workpiece. Moving the workpiece toward the liquid electrode initiates a spark discharge when the gap meets the breakdown condition and produces a high temperature to melt and erode the workpiece material. Similar to conventional μEDM, the liquid-metal electrode material will also be consumed, but liquid metal is continuously supplied to the nozzle tip to compensate for the consumption at the right pressure, thereby eliminating its impact on the removal process. In the current method, the system is configured so that the discharge pulses are generated only between the liquid electrode and workpiece through the controlled supply of liquid, and the capillary itself does not participate in the discharge; therefore, it remains intact. Thus, this method uses a liquid metal tip as a microdischarge electrode, which can be automatically compensated. Furthermore, its geometric shape can be controlled and constrained, which is highly desirable for microsized thin-walled flexible devices, thin-film sensors, or workpiece surface etching.

## 3. Shape of the Liquid-Phase Electrode

### 3.1. Mathematical Model

To obtain a liquid-metal electrode that meets the discharge demand, the formation conditions and morphology of the electrode are extremely critical.

We know that the surface tension at any point on the plane of the static liquid surface is the same in all directions, counteracting each other. In other words, the pressure just outside the surface *P*_o_ and just inside the surface *P_i_* is equal. Therefore, there is no additional pressure on the plane of the static liquid surface. However, if the pressure inside the surface *P_i_* is greater than the pressure outside the surface *P*_o_, an additional pressure *P_f_* will be generated and induce a curved liquid surface (if the drop is small, the effect of gravity may be neglected and the shape may be assumed to be spherical), and vice versa. Figure 2 shows that the tip of the liquid-metal electrode appears to be a convex sphere projected over the flat bottom of the coated needle (the liquid metal does not wet the coated needle).

From the geometric relationship of the solid–liquid contact surface shown in Figure 2, it can be found that the relationship between *θ* and *α* is
(1)$$\theta =\frac{\pi }{2}+\alpha ,$$
where *θ* is the liquid–solid contact angle (°), and *α* is the angle between the boundary radius of the curved liquid surface and the axis of the capillary (°).

The additional pressure at the tip of the liquid electrode is as follows [29]:(2)$$P_{f}=\frac{2\pi r^{2}\delta }{R}/\pi r^{2}=\frac{2\delta }{R}=\frac{2\delta \sin\left(\theta -\frac{\pi }{2}\right)}{r},$$
where δ is the coefficient of surface tension (N/m), *r* is the inner radius of the capillary (m), and *R* is the radius of the curved liquid surface (m).

There is a trigonometric relationship between the contact angle *θ* and the additional pressure *P_f_* produced by the curved liquid surface. When the contact angle is *θ* = π/2, the additional pressure is *P_f_* = 0 Pa. When the contact angle is *θ* < π/2, the additional pressure is *P_f_* < 0 Pa. This result indicates a negative pressure. The liquid surface at the tip of the liquid metal is concave, and the direction of additional pressure is downward, which will not happen in our research case. When the contact angle is *θ* = π, the additional pressure *P_f_* is at the maximum, and this state usually leads to a liquid flow or the ejection of liquid.

However, under natural conditions, the *P_f_* is too small, and the radius *R* of the convex sphere suspended at the end of the coated needle tip is too large. This result leads to the sagittal *h* (the height of a segment) being too small; thus, the protruding convex sphere used as an electrode cannot meet the requirement for the μEDM process. Therefore, extra pressure is applied to the needle. The pressure on the tip of the liquid-metal electrode is shown in Figure 3.

From Figure 3, the liquid metal remains stationary inside the needle at equilibrium, the same as the tip of the liquid metal. The additional pressure *P_f_* generated on the convex liquid surface of the liquid metal is
(3)$$P_{f}=P_{1}+P_{l1}-P_{l2},$$
where *P*_1_ is the extra applied pressure (Pa), *P*_l1_ is the liquid-metal gravity (Pa), and *P*_l2_ is the dielectric-liquid gravity (Pa).

Hence, the tip of the liquid-metal electrode will maintain its shape in the equilibrium state. Then, the contact angle *θ* of the liquid-metal electrode tip is
(4)$$\theta =\frac{\pi }{2}+\arcsin\left[\frac{\left(P_{1}+P_{l1}-P_{l2}\right)\cdot r}{2\delta }\right].$$

During the EDM process, a pulse generator is applied to the electrode and workpiece. This applied voltage will generate an electric field between the liquid-metal electrode and the workpiece. The electric field will exert an electric field force at the tip of the liquid-metal electrode, which will affect or even change the tip shape of the liquid-metal electrode. At the same time, the change in the tip shape of the liquid-metal electrode will change the electric field distribution between the electrode and the workpiece, and the electric field force on the tip of the electrode will also change.

In the case without an electric field force, as shown in Figure 4, the tip shape of the liquid-metal electrode will form a spherical shape due to surface tension, as discussed before.

Suppose that the liquid metal is the ideal conductor (its resistivity is assumed to be 0 (as conductivity approaches infinity), which makes calculations easy to perform), the surface of the liquid metal is smooth, and the electric field is an irrotational field; then, the electric field line at point A follows arc $\overset{⌢}{AB}$ and line BC. Suppose line segment DA is equal to *r*_0_; then, according to the geometric relationship in Figure 4, the length of line segment BC can be obtained as follows:(5)$$d_{1}=d_{0}-R(1-\cos\alpha ).$$

The length of arc $\overset{⌢}{AB}$ can be calculated as follows:(6)$$r_{0}\beta =R\tan(\frac{\beta }{2})\beta =\beta R\frac{1-\cos\beta }{\sin\beta }.$$

The relationship between the electric potential *U* at point A and the electric field intensity *E* can be expressed as (7)$$U=\int_{\overset{⌢}{AB}+BC}\vec{E}d\vec{l}=E(r_{0}\beta +d_{1})=E(\frac{\beta R(1-\cos\beta )+d_{1}\sin\beta }{\sin\beta }).$$

Then, the average electric field intensity *E* on line ABC can be obtained as follows:(8)$$E=\frac{U\sin\beta }{\beta R(1-\cos\beta )+d_{1}\sin\beta }.$$

In the spherical coordinate system with point *O* as the center of the ball and *R* as the radius, by taking an area differential element *ds* near point A, the charge of point A in this differential element can be calculated as
(9)$$dq=\varepsilon Eds=\frac{\varepsilon U\sin\beta }{\beta R(1-\cos\beta )+d_{1}\sin\beta }R^{2}\sin\beta d\beta d\gamma ,$$
where *γ* is the azimuth in the coordinate system with a range of 0–2π, and *ε* is the dielectric constant.

Integrating the entire convex spherical surface, the charge of the entire spherical surface can be obtained as follows:(10)$$\begin{array}{ll} q & =\int_{0}^{2\pi }\int_{0}^{\alpha }\frac{\varepsilon U\sin\beta }{\beta R(1-\cos\beta )+d_{1}\sin\beta }R^{2}\sin\beta d\beta d\gamma \\ & =2\pi \varepsilon UR^{2}\int_{0}^{\alpha }\frac{\sin^{2}\beta }{\beta R(1-\cos\beta )+d_{1}\sin\beta }d\beta . \end{array}$$

According to the principle of virtual displacement, the electric field force received by the convex spherical surface is given as
(11)$$\begin{array}{ll} Fe & =\frac{1}{2}\frac{\partial (q/U)}{\partial d_{1}}U^{2}\\ & =\pi \varepsilon U^{2}R^{2}\int_{0}^{\alpha }\frac{\sin^{3}\beta }{{\left[\beta R(1-\cos\beta )+d_{1}\sin\beta \right]}^{2}}d\beta . \end{array}$$

Since *β* < π/2, *β* can be approximately replaced by 2 sin(*β*/2), substituting this into Equation (11) provides the following:(12)$$\begin{array}{ll} Fe & =\frac{\pi \varepsilon U^{2}}{2}[\frac{2(d_{1}+R)}{d_{1}}+\frac{2R(d_{1}-4R)}{16R^{2}+d_{1}^{2}}+\ln\frac{d_{1}}{2R\sin^{2}(\alpha /2)+d_{1}\cos(\alpha /2)}\\ & +\frac{4R(8R^{2}+d_{1}^{2})+2d_{1}(12R^{2}+d_{1}^{2})\cos(\alpha /2)}{\left[R\left(\cos\alpha -1\right)-d_{1}\cos(\alpha /2)\right]\left(16R^{2}+d_{1}^{2}\right)}\\ & -\frac{d_{1}(24R^{2}+d_{1}^{2})}{{\left(16R^{2}+d_{1}^{2}\right)}^{3/2}}\ln\frac{{\left[\left(\sqrt{16R^{2}+d_{1}^{2}}+4R\right)\left[1-\cos(\alpha /2)\right]+d_{1}\left[1+\cos(\alpha /2)\right]\right]}^{2}}{2R\sin^{2}(\alpha /2)+d_{1}\cos(\alpha /2)}. \end{array}$$

Equation (12) shows the electric field force of the entire liquid-metal convex sphere. This is to analyze the entire spherical surface as a whole. However, to analyze the influence of the electric field force on the tip shape of the liquid-metal electrode, it is necessary to divide the spherical surface of the liquid-metal tip into an infinite number of microelements to analyze the force situation of each microelement. The electric field force near any point A on the spherical surface is
(13)$$dF_{e}=\frac{1}{2}\frac{\varepsilon U^{2}R^{2}\sin^{3}\beta }{{\left[\beta R(1-\cos\beta )+d_{1}\sin\beta \right]}^{2}}d\beta d\gamma .$$

If *r* = 10 µm, *d*_0_ = 100 µm, and *U* = 100 V, the value range of *β* is [−π/2–π/2], and *dβ* and *dγ* are constant at 1; then, the curve of the electric field force *dFe* around *β* at any point A on the spherical surface is as shown in Figure 5.

As shown in Figure 5, the electric field force on the spherical surface is not evenly distributed. The electric field force is the largest at the lowest point G of the spherical surface, and, with point G as the center, the electric field force gradually decreases toward point E, which is similar to point F. This nonuniform distribution of the electric field force may cause the tip shape of the liquid-metal electrode to change from a spherical shape to a Taylor cone.

### 3.2. Simulation

#### 3.2.1. Geometric Model and Boundary Condition

Figure 6 shows the geometric size and boundary conditions of the model used for simulation. The size of the simulation area is 0.4 × 0.6 mm, the length of the capillary nozzle is 0.5 mm, and the distance between the nozzle tip and the workpiece is 0.1 mm. Zone A is the liquid metal inside the electrode, zone B is stainless steel, zone C is the EDM oil, and zone D is a parylene C coating with a thickness of 20 μm, as shown in Table 1, where *E* the electrostatic field on the boundary when a voltage is applied between liquid-metal electrode and workpiece.

At present, most metals or alloys are in the solid state at room temperature. Exceptions include francium, cesium, rubidium, mercury, sodium–potassium alloys, and gallium-based alloys, which can be defined as liquid metals. Their melting points are either lower than or close to room temperature, which enable them to remain in the liquid state at room temperature. Unfortunately, the intrinsic radioactivity of cesium, extreme instability of francium and rubidium, flammability and corrosivity of sodium–potassium alloys, and toxicity of mercury limit their applications to certain specific areas. On the other hand, gallium-based alloys, such as Galinstan (68.5 wt.% gallium, 21.5 wt.% indium, and 10 wt.% tin), a commercially available eutectic liquid alloy, is a low-activity and nontoxic liquid with a low melting point (−19 °C) and low viscosity (0.0024 Pa·s at 20 °C) that allows it to be easily transferred through microscale nozzles. Its high electrical conductivity (~3.5 × 10^6^ S/m at 20 °C), high thermal conductivity (16.5 W·M^−1^·K^−1^), and high boiling point (>1300 °C) are desirable features for a μEDM electrode application. Therefore, Galinstan is used as the liquid electrode in this study.

#### 3.2.2. Tip Shape of the Liquid Electrode at Different Extra Pressures

From the theoretical analysis in Section 2, it is known that the extra pressure is an important factor affecting the contact angle *θ* of the liquid-metal electrode. To evaluate this effect in detail, the influence of extra pressure on the tip shape of the liquid-metal electrode was simulated by varying extra pressures. The simulation results are shown in Figure 7. When the extra pressure *P*_1_ is 0.1 atm, the liquid metal cannot be extruded from the needle (Figure 7a). The reason may be that the extra pressure is too small to overcome the internal frictional resistance caused by Galinstan’s viscosity. As the extra pressure increases, the liquid-metal electrode tip gradually becomes a cone (Figure 7b,c). A larger extra pressure results in a faster change in the tip shape of the liquid-metal electrode. The tip shape of the liquid-metal electrode gradually changes into a spherical shape under the action of pressure. The liquid metal gradually extends to the parylene coating at the bottom of the needle, making the radius of the liquid-metal electrode tip larger than the inner diameter of the needle. This result implies that the width of the groove patterning by liquid-metal electrode μEDM may be larger than the inner diameter of the needle. Moreover, as the extra pressure increases, the sagitta of the tip of the liquid-metal electrode grows longer (Figure 7d). This result verifies the analysis results of Section 2. It is worth noting that, for an extra pressure over 2 atm, the liquid metal will promptly eject and touch the workpiece (Figure 7e). Therefore, it is necessary to set an appropriate pressure to obtain an electrode that meets processing needs.

#### 3.2.3. Tip Shape of the Liquid Electrode at Different Voltages

Figure 8 shows the simulation result of the tip shape of the liquid-metal electrode at different voltages (tested up to 2 kV, with an extra pressure of *P*_1_ = 1.2 atm). As shown in Figure 8a,b, it can be seen that the tips of the liquid-metal electrode are almost the same in the case of low voltage. This outcome is reasonable because the electric force at low voltage is not high enough to affect the tip of the liquid-metal electrode.

When the voltage is 0.5 kV, as shown in Figure 8c, the tip of the liquid-metal electrode is gradually stretched into an ellipsoidal shape under the combined action of the electric field force and the extra pressure. Compared with Figure 8a or 8b, the sagitta of the tip shape of the liquid-metal electrode is even longer and changes more rapidly over time. This may be attributed to the tensile effect of the electric field force generated by the high voltage on the liquid metal. As the voltage continues to increase, the tip of the liquid-metal electrode will clearly change. As seen from Figure 8d,e, the tip of the liquid-metal electrode changes over time and is gradually stretched into an inverted cone. Compared with Figure 8c, the tip of the liquid-metal electrode is now more of a conical shape, with a smaller diameter. The electric field force generated by the voltage constrains the tip shape of the liquid-metal electrode, and the constraint becomes stronger with increasing voltage. When the electric field force generated by the additional high voltage is sufficiently large, the liquid-metal electrode terminal can be stretched into a reverse cone.

It is also worth noting that for a voltage greater than 2 kV, the tip of the liquid-metal electrode is gradually stretched into an inverted cone and then ejected to the workpiece (as shown in Figure 8f). This result can be explained as the stretching effect of the liquid metal under the extra pressure and electric field force. This explanation is reasonable because the electric field force is too strong.

From the simulation, it can be seen that the effect of a low voltage on the electrode tip is not obvious, but the effect of a high electric field force is very obvious. Therefore, the external electric field may be an appealing way to restrain the tip of the liquid-metal electrode in future research.

## 4. Experimental Set-Up and Procedure

The schematic in Figure 9 illustrates the μEDM process experiments. The set-up has a servo-controlled three-axis stage with a 100 nm positioning resolution, comprising an *XY* stage with the holder on which the work tank is held and a *Z* stage to vertically position the syringe containing the needle with a liquid-metal electrode. The work tank is configured to have a sample holder made of plastic, in which the workpieces are immersed in a dielectric EDM fluid, fixed and electrically coupled with the discharge circuit using conductive adhesive tape. The nozzle is connected to the syringe that stores Galinstan, and a pressurizing unit is used to apply pressure to the syringe to feed Galinstan to the needle. To prevent the oxidation of Galinstan, a low-concentration (2%) sulfuric acid solution is added to the syringe and floats on top of Galinstan. Furthermore, the H_2_SO_4_ solution keeps the liquid metal clean to avoid clogging the needle. Galinstan is electrically coupled with the discharge circuit using a conductive copper wire immersed in it and sealed with glue on the syringe wall.

The system employs a relaxation-type (resistor–capacitor, or RC) pulse generator with a variable direct current (DC) voltage source. In this type of pulse generation circuit, the loop-voltage equation can be expressed as
(14)$$U_{c}=E(1-e^{-\frac{-t}{T}}),$$
where *U*_c_ is the voltage on the capacitor C (V), *E* is the voltage of the DC source S (V), and *T* = *RC*. In this case, the discharge energy is stored in the capacitor. When the voltage of the capacitor meets the condition of *U_c_* ≥ *U_d_* (*U*_d_ is the breakdown voltage of the discharge gap), the gap discharges and instantly releases the energy forming a pulse current. The generated discharge pulses are monitored using a detection circuit, and the readouts of the signals are obtained via a General Purpose Interface Bus (GPIB) interface connected to a computer; moreover, a probe is coupled with the discharge circuit and connected to an oscilloscope for visual observations.

A stainless-steel needle (outer diameter *D*_O_ = 240 μm, inner diameter *D_I_* = 100 μm, length *L* = 13 mm) is used as the needle in the set-up. The needle is coated with a dielectric film, parylene C (thickness, 20 μm), to eliminate the effects of discharge on the needle surface. The coating consequently modifies the *D*_O_ and *D_I_* of the needle to 280 and 60 μm, respectively. Although the distance, *D*, between the workpiece surface and the needle tip cannot be directly determined via electrical surface detection, it can make use of a nozzle with the presence of Galinstan at its tip. Briefly, a low pressure (less than 0.5 atm) is applied via a precision pressure device to ensure that the liquid metal in the needle is just flush with the tip of the needle. Then, using a small voltage (10 V) between them electrical surface detection is carried out using the *Z*-axis of the stage at a low speed, and retracting the needle (50 μm) to determine the gap, as shown in Figure 10a–c. To decrease the measuring error, electrical surface detection both toward and away from the gap is repeated continuously three times, and the average is taken as the gap position. The μEDM process with this setting is performed in the die-sinking mode, and the workpiece is electrically coupled with the discharge circuit cathode while the electrode is connected to the anode. After applying a normal pressure at the nozzle and applying a machining voltage between the nozzle and workpiece, the gap *d* between the liquid metal and the workpiece is adaptively determined during discharging, with horizontal scanning of the workpieces using the *XY* stage with respect to the liquid electrode having a constant gap distance *D* for the lateral removal of the material using the liquid electrode, as shown in Figure 10d. The experimental conditions are outlined in Table 2.

Machining characteristics such as the material removal rate (MRR) (mm^3^/min) and liquid-metal consumption rate (LMCR) (mm^3^/min) are adopted to evaluate the effects of the machining parameters on the liquid-metal electrode μEDM processes. The MRR is computed as the ratio of the material removed from the workpiece (approximated as the volume of the frustum of the cone) to the recorded machining time of the EDM system. The LMCR is calculated as the ratio of the liquid metal consumed to the machining time.

## 5. Experimental Results and Discussion

### 5.1. Microgroove and Arbitrary Patterning

Microgroove erosion was performed through programming the motions of the three-axis stage. An example of a microgroove pattern using the coated needle is shown in Figure 11. The electrical conditions were *C* = 27 nF, *R* = 1 kΩ, *V_s_* = 100 V, and *P*_1_ = 1.2 atm. The tool scanning path in the *XY* plane was repeated for 200 scans at scanning speeds of up to 1.0 mm/s with a constant gap distance of *D* = 100 µm. A pressure of 1.2 atm was applied to the syringe. The profile of the produced groove was analyzed using a laser confocal microscope, and the sample results are included in Figure 12. The linewidth and depth of the produced groove were measured to be approximately 280 and 90 μm, respectively. The cross-sectional profile of the groove was similar to the shape of the liquid electrode tip. This result is reasonable because EDM is a copying process mode, and the groove will match the electrode. In addition, randomly distributed pits appeared at the edge of the processed groove in Figure 11a. The reason for the micropits near the edges of the produced groove is that some pulses were likely generated between the suspended liquid microelectrode and debris comprising the removed material of the workpiece and the liquid metal that had already dripped on the workpiece surface. By programming the contour of the nozzle motion with respect to the *XY* plane (with the same *D* value), arbitrary patterning was successfully demonstrated on the silicon sample (Figure 12).

### 5.2. Process Characterizations

#### 5.2.1. Dependence of Machining Characteristics on the Pressure

The widths of the linear patterns produced by lateral scanning of the Galinstan electrode were characterized while varying the pressure. The line widths of the produced patterns were measured using an optical measuring microscope. Figure 13 shows the relationship between the measured line widths and pressure (with *V_s_* = 100 V, *C* = 100 nF, *R* = 0.5 kΩ, as shown in Table 2). As shown in the graph, an optimal pressure value for the minimum line width was found under the same discharge conditions.

When the applied pressure was small, the liquid-solid contact angle *θ* was also small; hence, the tip of the liquid electrode was large. With an increase in applied pressure, the contact angle *θ* gradually increased, and the radius *r* of the curved liquid surface gradually decreased. That is, the end of the electrode correspondingly decreased; thus, the width of the machined groove decreased directly. When the applied pressure continued to increase, although the contact angle *θ* gradually increased, the radius *r* of the curved liquid surface gradually decreased, forming an electrode with a finer end; however, the width of the groove gradually increased. The reason for this result may be that when the applied pressure is greater, the liquid metal at the tip is more likely to fall onto the workpiece, which will affect the processing and cause discharge instability. Moreover, we also found that, when the external pressure was greater than 2 atm, the liquid metal ejected like a continuous stream of water; thus, it was impossible to carry out EDM.

In this connection, it is clear that there was a similar trend in the relationship between the MRR and pressure *P*_1_, as shown in Figure 14. During the discharge process, when the pressure was small, the consumed Galinstan could not be compensated in time, leading to an open-circuit state most of the time; this resulted in the small value of the MRR. The MRR increases with increasing pressure because the frequency of discharge increases with an increasing Galinstan compensation. After the optimal pressure value, as the pressure increases, excess Galinstan is suspended on the needle tip, and this droplet easily falls due to the discharge and electrostatic forces and causes unstable discharge, thus resulting in the MRR decreasing.

Figure 15 shows the relationship between the LMCR and pressure *P*_1_. As shown in the graph, the LMCR increased with increasing pressure in a consistent manner. This outcome is reasonable because the flow rate, which determines the consumption of Galinstan based on the velocity of the flow, increased with increasing pressure. It is worth noting that, for pressures over 1.3 atm, the amount of fed Galinstan was much greater than the amount of Galinstan consumed by discharge at an excessive flow velocity. This result is disadvantageous for both the MRR and LMCR. Therefore, the feeding of consumed Galinstan is very important for high efficiency and stable processing, which largely depends on the applied pressure.

#### 5.2.2. Dependence of Machining Characteristics on the Voltage

Figure 16 plots the measured line widths as a function of the voltage (tested up to 220 V, with constant *C* = 100 nF, *R* = 0.5 kΩ, and *P*_1_ = 1.3 atm, as shown in Table 2), and the figure clearly shows that the line width increased with the applied voltage in a consistent manner. This result is reasonable as the discharge energy, which determines the amount of material removed by a single pulse, increased with the voltage. In this regard, as shown in Figure 17 and Figure 18, a similar trend was obtained for both the MRR and LMCR by varying the voltage *V*_s_; the trends were consistent given the dependence of the discharge energy on the voltage. Increasing the discharge energy caused a greater spark intensity, which generated more melted material in the spark region. Thus, the workpiece and the electrode material were both subjected to an increase in sparks, which increased both the MRR and the LMCR. As a result, the MRR and LMCR substantially increased with increasing discharge energy.

It can be seen from the simulation that a higher voltage resulted in a greater electric field, whereas, a larger contact angle *θ* resulted in a smaller radius *r* of the curved liquid surface. That is, the electrode tip was correspondingly smaller, and the widths of the linear patterns produced by the lateral scanning of the Galinstan electrode should be smaller as the voltage increases. However, the line width showed the opposite trend in this experiment, in that the line width increased with the voltage. There may be two reasons. First, the material removal depends on the energy of a single electric discharge, *E*_SED_, which is expressed as
(15)$$E_{SED}=\frac{1}{2}CU^{2},$$
where *C* is the capacitance of the RC circuit and *U* is the machining voltage. From Equation (15), it can be seen that, as the voltage increases, the discharge energy of a single pulse increases squarely with the voltage; thus, the amount of removed material increases. The effect of increasing the voltage on the energy for material removal is much greater than that on the size of the electrode.

Second, the applied voltage for μEDM is relatively small, and the voltage changes within a small range of 50 to 220 V. The size of the electrode tip does not change significantly at low voltages. Only when the voltage is over 1 kV will a change in the liquid electrode tip be apparent. However, it is impossible to use such a high voltage for μEDM.

Therefore, changing the size of the electrode by increasing the voltage shows a much smaller effect on groove width than the effect of increased energy by increasing the voltage. Notably, from an energy point of view, an increase in voltage is beneficial to the MRR.

## 6. Conclusions

In this paper, the shape of the electrode, which is used in liquid-electrode μEDM processing for the purpose of resolving the problems related to electrode wear in traditional μEDM, was analyzed and simulated. Moreover, experiments were performed for verification. Arbitrary patterns on silicon samples using a liquid-metal electrode were well presented. The influence of extra pressure and open voltage *V_s_* on the line widths, MRR, and LMCR were also investigated. The external pressure had a significant impact on the tip shape of the liquid electrode and its compensation. There was an optimum pressure value for the minimum line widths and maximum MRR. The LMCR increased with pressure; hence, an appropriate pressure was also beneficial to the LMCR. Although the voltage also influenced the tip shape of the liquid electrode, it was negligible at low voltage. However, at low voltages, the line widths, MRR, and LMCR consistently exhibited the same trend as the voltage. That is, they increased with voltage. Thus, the single-pulse discharge energy, which determines the amount of material removed by a single pulse, increased with the voltage.

## Figures and Tables

**Figure 1 micromachines-11-00935-f001:**
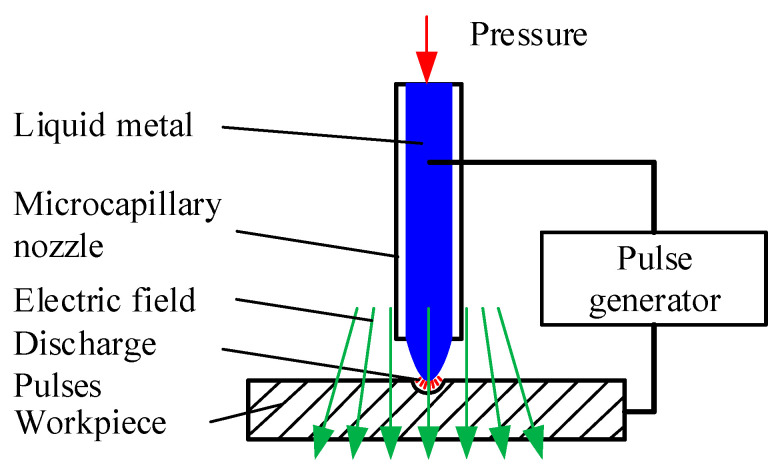
Principle of micro-electro-discharge machining (μEDM) using a liquid-metal electrode.

**Figure 2 micromachines-11-00935-f002:**
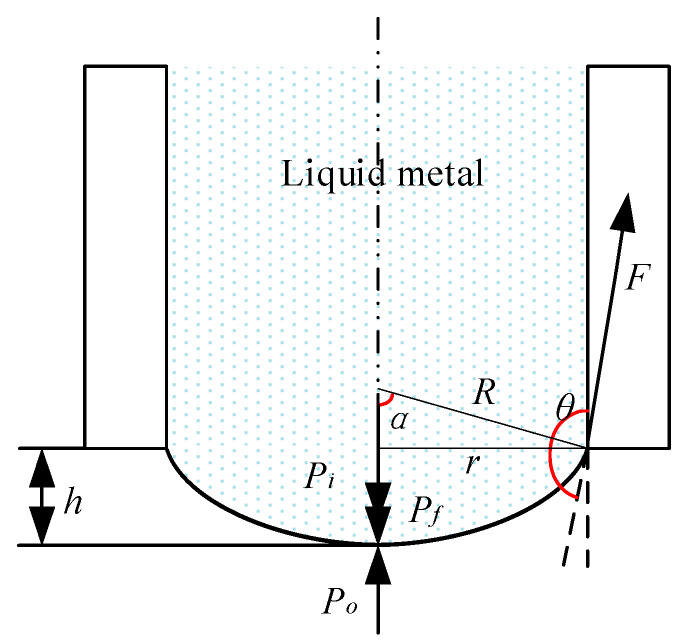
Tip shape of the liquid-metal electrode.

**Figure 3 micromachines-11-00935-f003:**
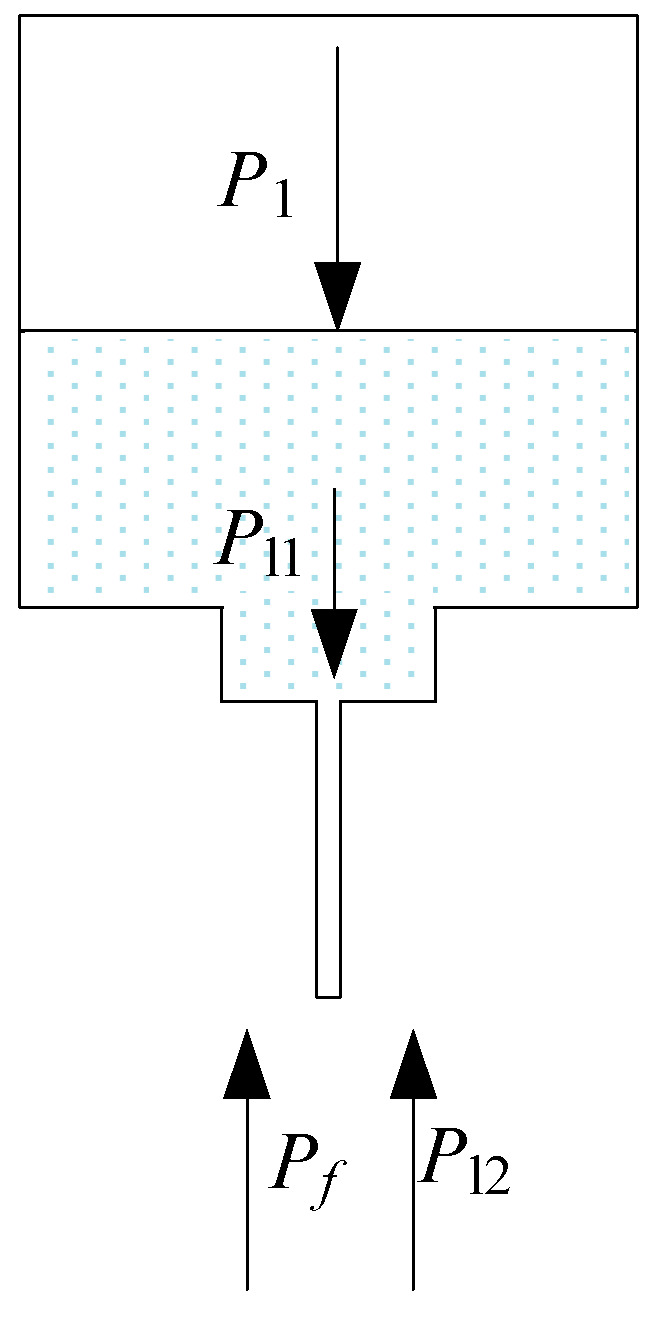
Pressures of the liquid-metal electrode on the nozzle tip.

**Figure 4 micromachines-11-00935-f004:**
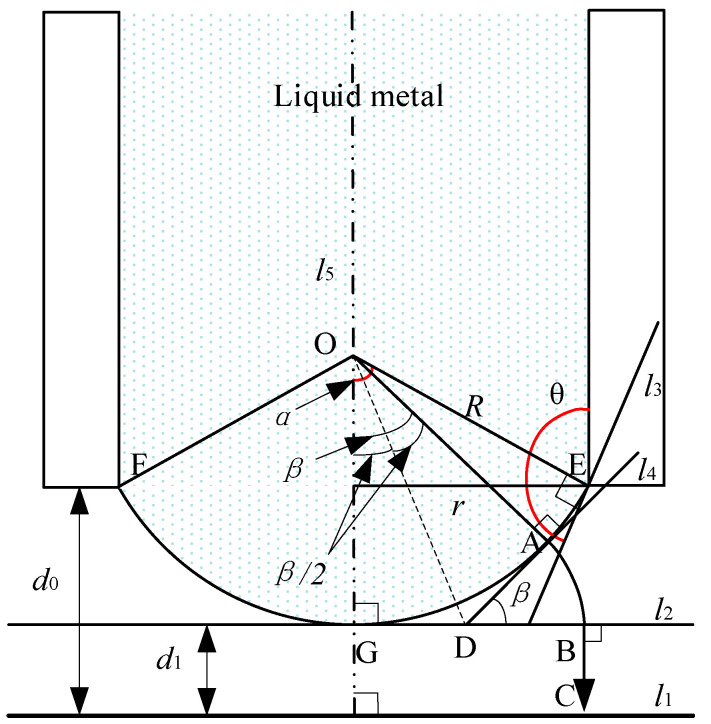
Electric field force of the liquid-metal electrode tip.

**Figure 5 micromachines-11-00935-f005:**
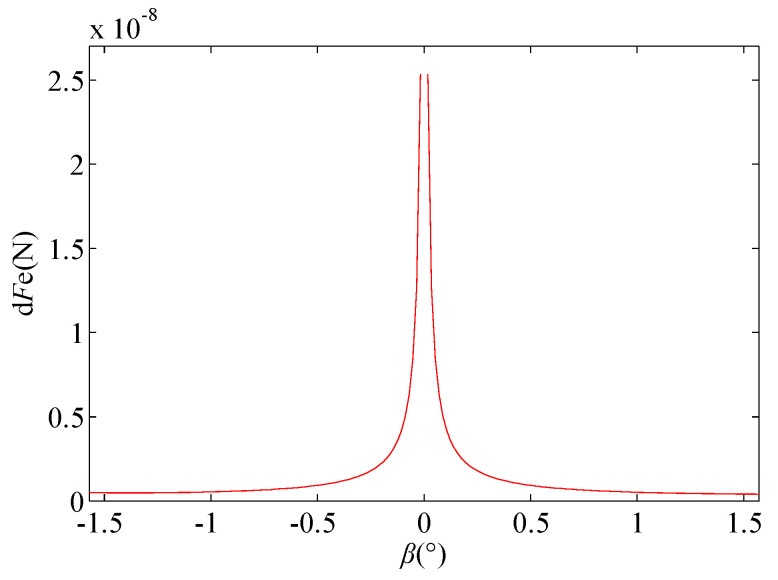
Relationship between the electric field force *dFe* and *β* at any point A on the spherical surface.

**Figure 6 micromachines-11-00935-f006:**
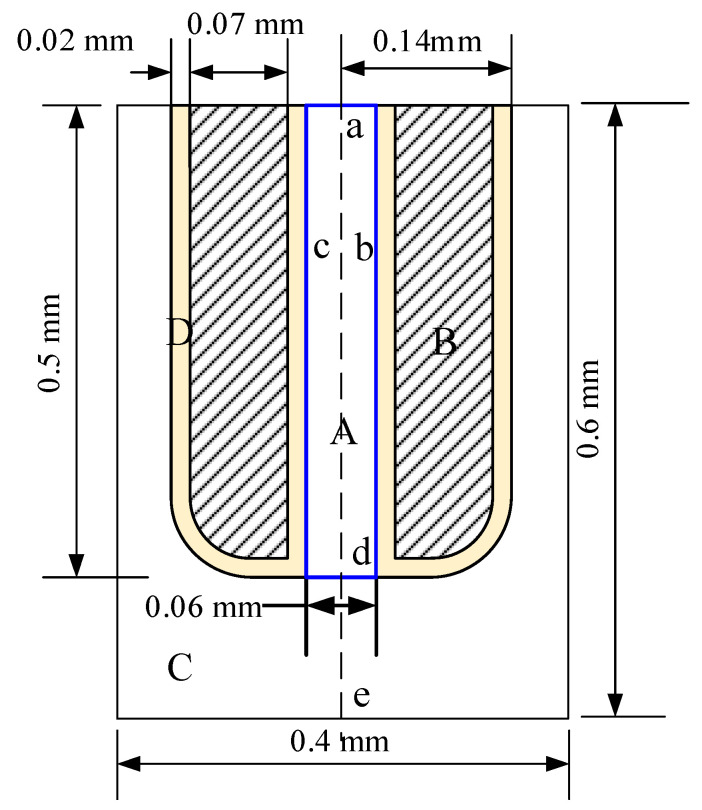
Geometric size and the boundary conditions of the model.

**Figure 7 micromachines-11-00935-f007:**
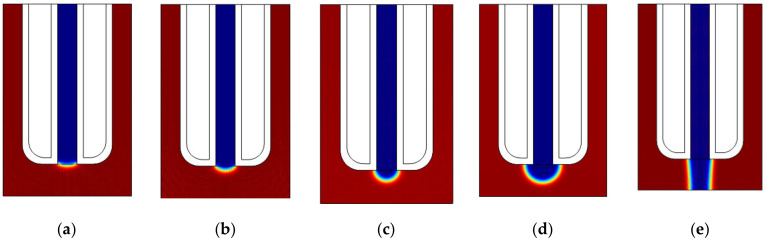
Simulation results at different extra pressures. (**a**) 0.1 atm. (**b**) 0.8 atm. (**c**) 1.2 atm. (**d**) 1.4 atm. (**e**) 2 atm.

**Figure 8 micromachines-11-00935-f008:**
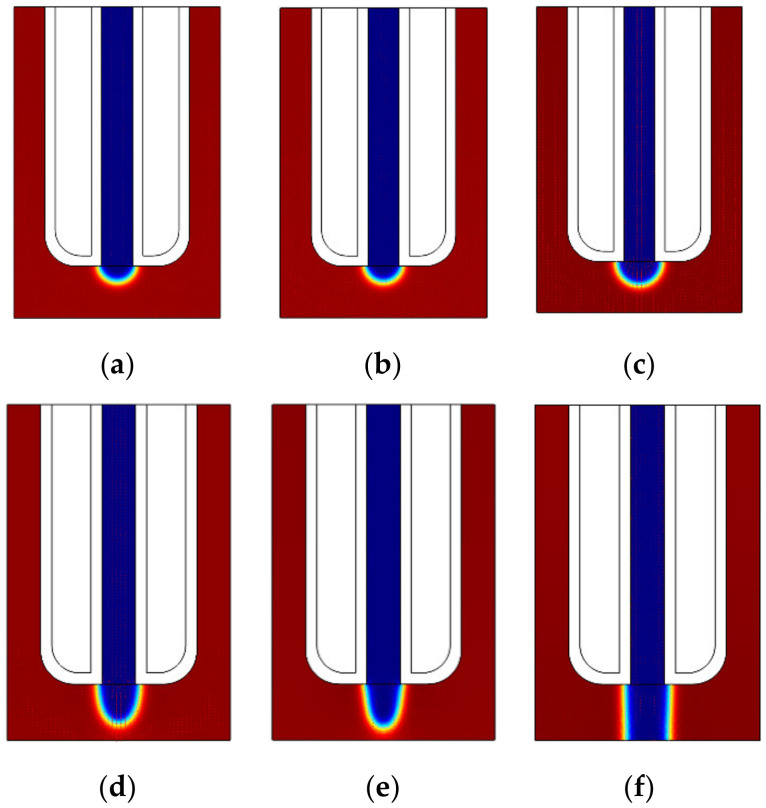
Simulation results at different voltages. (**a**) 50 V. (**b**) 100 V. (**c**) 0.5 kV. (**d**) 1 kV. (**e**) 1.5 kV. (**f**) 2 kV.

**Figure 9 micromachines-11-00935-f009:**
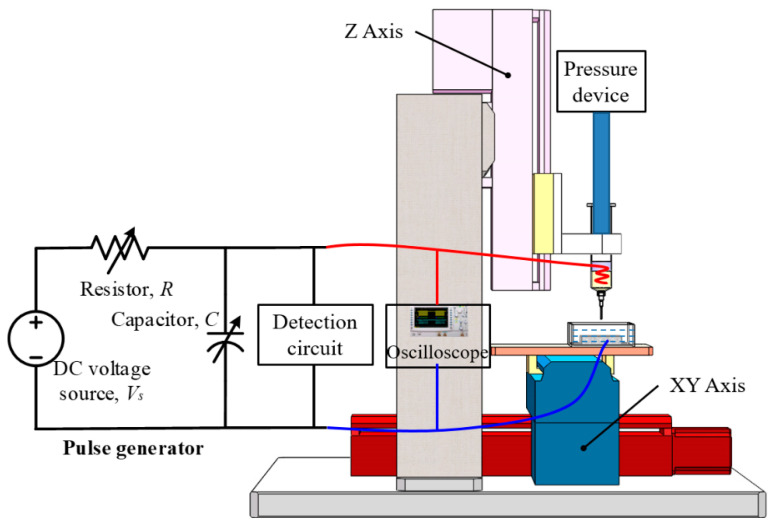
Experimental set-up for liquid-metal electrode μEDM.

**Figure 10 micromachines-11-00935-f010:**
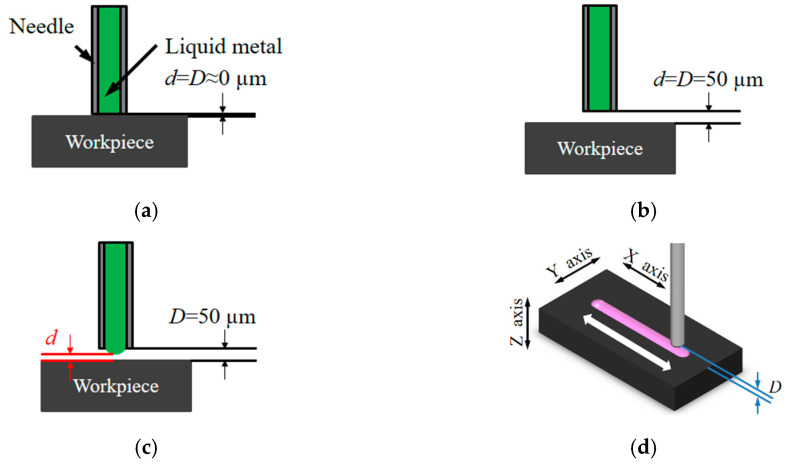
Electrical surface detection and scanning erosion. A low pressure (0.1 atm) is applied via a precision pressure device to ensure that the liquid metal in the needle is just flush with the tip of the needle; then, a small voltage (10 V) is used between them to carry out electrical surface detection using the *Z*-axis of the stage at a low speed from (**a**) *D* = 0 μm (**b**) to *D* = 50 μm (retracting the needle) to determine the gap. (**c**) The above electrical surface detection is repeated three times, and the average is taken as the gap position. Additionally, the gap *d* between the liquid metal and the workpiece is adaptively determined during discharging. (**d**) Horizontal scanning of the workpieces using the *XY* stage with respect to the liquid electrode at a constant gap distance *D* for the lateral removal of the material using the liquid electrode.

**Figure 11 micromachines-11-00935-f011:**
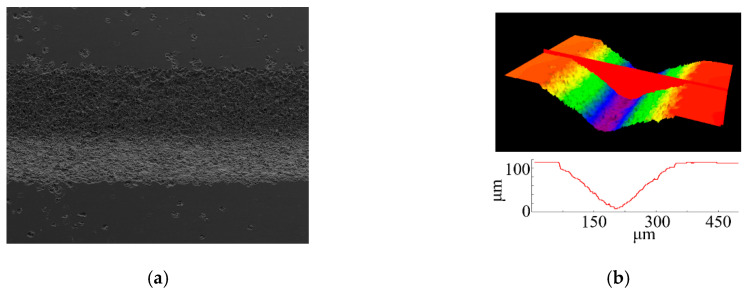
Linear groove pattern. (**a**) Sample of the microgroove and (**b**) profile of the microgroove.

**Figure 12 micromachines-11-00935-f012:**
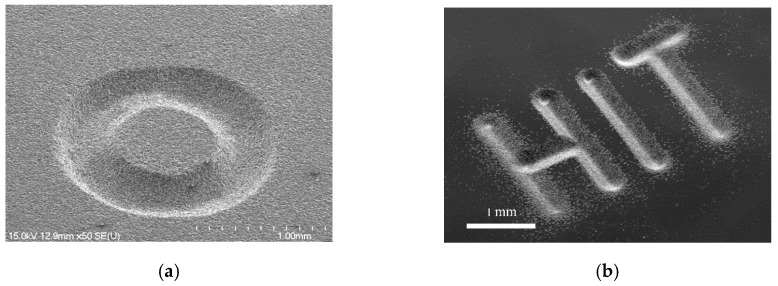
Sample result of arbitrary scanning-mode patterning performed using the developed process: (**a**) convex circular pillar and (**b**) “HIT” characters.

**Figure 13 micromachines-11-00935-f013:**
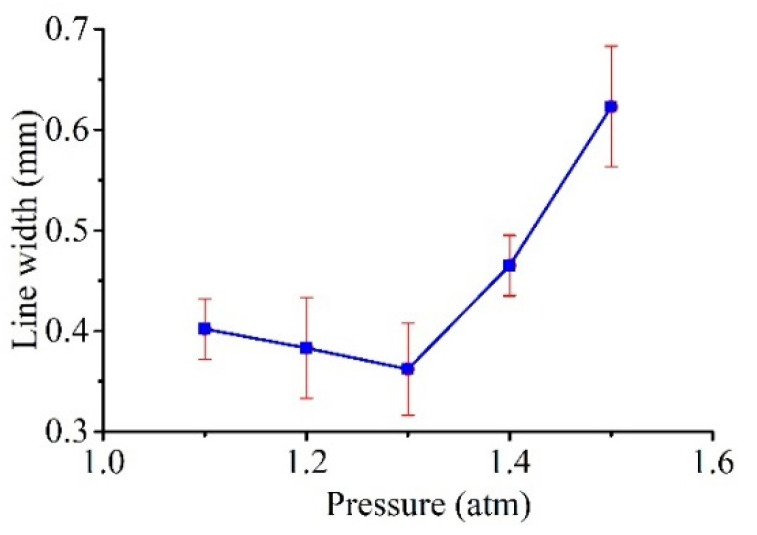
Effect of pressure *P*_1_ on the line widths.

**Figure 14 micromachines-11-00935-f014:**
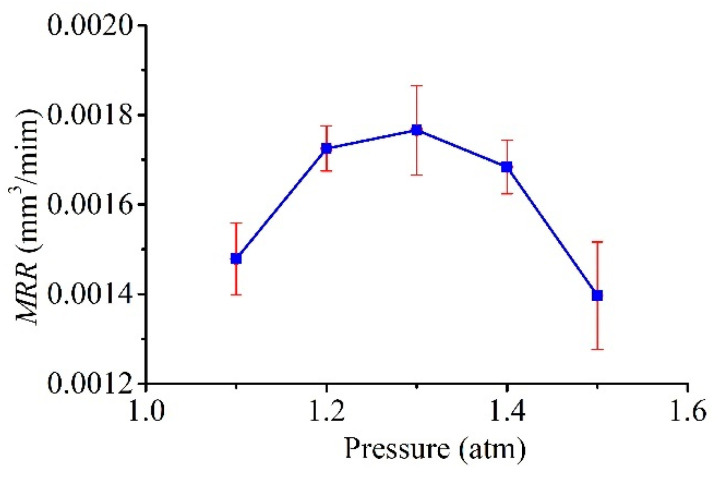
Effect of pressure *P*_1_ on the material removal rate (MRR).

**Figure 15 micromachines-11-00935-f015:**
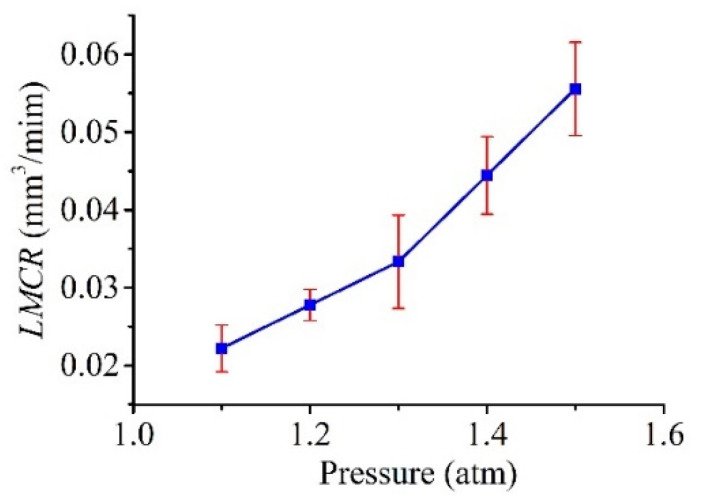
Effect of pressure *P*_1_ on the liquid-metal consumption rate (LMCR).

**Figure 16 micromachines-11-00935-f016:**
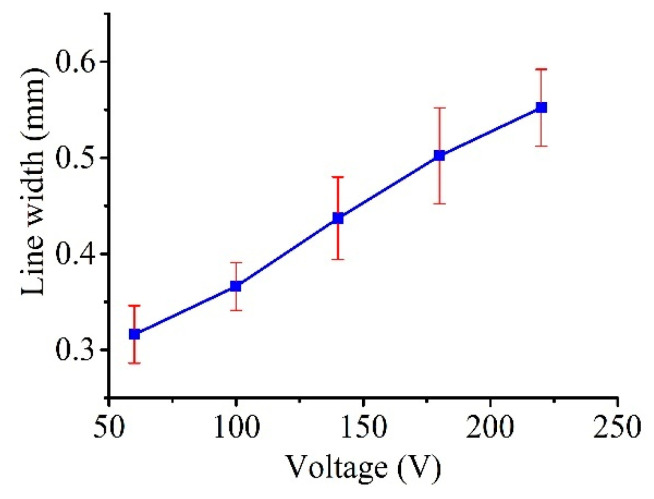
Effect of voltage *V_s_* on the line widths.

**Figure 17 micromachines-11-00935-f017:**
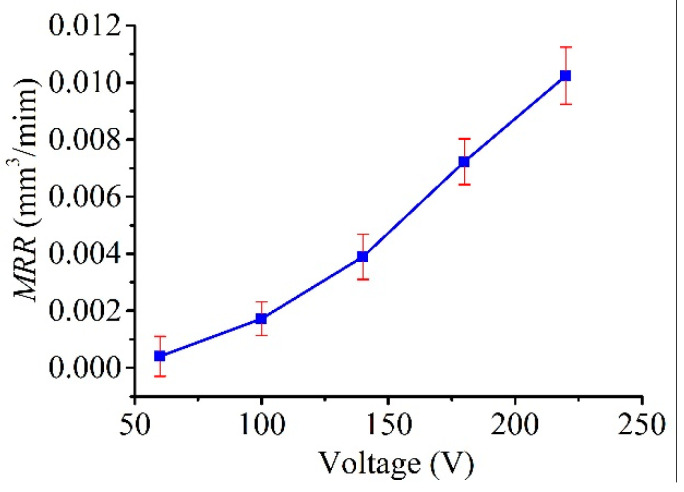
Effect of open voltage *V_s_* on the MRR.

**Figure 18 micromachines-11-00935-f018:**
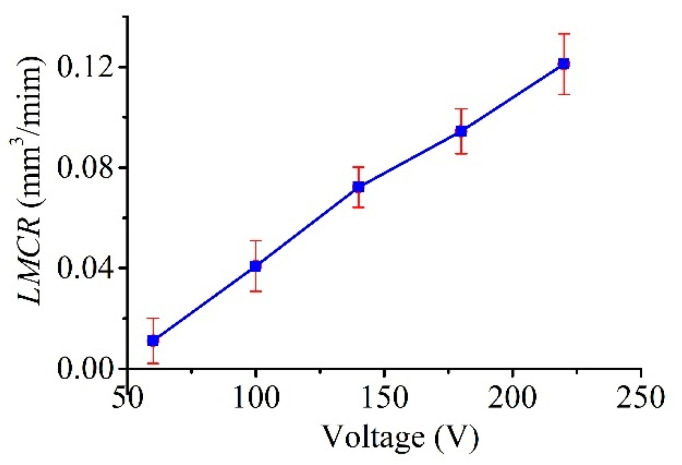
Effect of voltage *V_s_* on the LMCR.

**Table 1 micromachines-11-00935-t001:** Boundary conditions of the model.

Boundary	Fluid Field	Electrostatic Field
a	Entrance, extra pressure *P*_1_	*E*
b,c	No slip wall	*E*
d	Initial interface	*E*
e	Export	0

**Table 2 micromachines-11-00935-t002:** Experimental conditions.

Parameters	Conditions
Workpiece	Doped silicon (*p* type)
Voltage (*V_s_*)	60–220 V
Resistance (*R*)	0.2–10 kΩ
Capacitance (*C*)	10–470 nF
Pressure (*P*_1_)	1.1–1.5 atm
Working medium	Kerosene
Lateral scan speed	0.5 mm/s

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
