# Peer review of "Investigation of a Liquid-Phase Electrode for Micro-Electro-Discharge Machining"

_micromachines, 2020, doi:10.3390/mi11100935_

Round 1

Reviewer 1 Report

The manuscript investigates the capability of using a liquid-metal electrode for micro- electro-discharge machining. The manuscript is nicely written, clear and appealing. The information is relevant and the results are very interesting.

My comments are as follow

The introduction is clear and cover many aspects of the study motivation with relative references. However, the main aspect related to the liquid-metal electrode is not covered by reference. There are at least two works by the first two author that can be cited and discussed:

  • Huang, R., Xiong, X., Yi, Y. et al.Liquid alloy electrode for no-wear micro electrical discharge machining. Int J Adv Manuf Technol 106, 1281–1290 (2020). https://doi.org/10.1007/s00170-019-04693-z
  • Huang RN, Yi Y, YuWB, Takahata K (2018) Liquid-phase alloy as a microfluidic electrode for micro-electro-discharge patterning. J Mater Process Tech 258:1–8. Https://doi.org/10.1016/j.jmatprotec.2018.03.012

The liquid-metal used in the study is nominated only in raw 281 in the experimental setup and procedure section. I can suppose that the same material has been adopted for the simulation of section 2.2. In this case, if the material properties have been implemented in the numerical model, they should be reported in the section.

The author should also discuss they choice of the liquid metal.

In 227- 229, the manuscript reports: “When the extra pressure P1 is 0.1 atm, the liquid metal cannot be extruded from the needle because the extra pressure is too small to overcome the frictional resistance of the capillary inner wall”. How the frictional resistance is modelled in the model? Is it referred to the “no slip wall” condition? If yes, is this condition confirmed by experiments?

Please, check the figure 8, where probably there is a picture over the Fig8(f)

About the process.

Is the dielectric fluid in a static condition? In case the kerosene flows between electrode and tool, have the authors found some influence on tool shape?

It would be of great interest to evaluate the surface quality of the machined workpiece.

Since the needle must not contribute to the removal process, I suppose that there is a limitation in machining depth that cannot exceed the “h” parameter of fig 2. Moreover, as clearly exposed in the manuscript, the tool electrode shape is assimilable to a cone. This shape does not allow to produce straight wall for the machined geometry. Therefore, even if the presented approach solves the tool wear problem, it seems that there are strong limitations for its adoption. Can the authors list some real case in which the method has been applied?

Author Response

October 8, 2020

Dear Reviewer:

Thank you very much for giving us an opportunity to revise our manuscript. We very much appreciate your efforts to provide constructive comments and suggestions on our manuscript entitled “Investigation of a liquid-phase electrode for micro-electro-discharge machining”.

We have studied your comments carefully. According to your detailed suggestions, we have made a careful revision on the original manuscript. We would like to submit the revised manuscript for your kind consideration. Thank you very much.

The response to your comments are listed as follows.

  • The manuscript investigates the capability of using a liquid-metal electrode for micro- electro-discharge machining. The manuscript is nicely written, clear and appealing. The information is relevant and the results are very interesting.

Thank you very much for your interest in our work.

  • The introduction is clear and cover many aspects of the study motivation with relative references. However, the main aspect related to the liquid-metal electrode is not covered by reference. There are at least two works by the first two author that can be cited and discussed:

Huang, R., Xiong, X., Yi, Y. et al. Liquid alloy electrode for no-wear micro electrical discharge machining. Int J Adv Manuf Technol 106, 1281–1290 (2020). https://doi.org/10.1007/s00170-019-04693-z

Huang RN, Yi Y, YuWB, Takahata K (2018) Liquid-phase alloy as a microfluidic electrode for micro-electro-discharge patterning. J Mater Process Tech 258:1–8. Https://doi.org/10.1016/j.jmatprotec.2018.03.012

Thank you very much for your suggestions. We will cite and discuss our work in the introduction as follows.

“To address the wear-related problems associated with μEDM, Huang et al. proposed a novel μEDM method that uses a liquid alloy as the machining electrode instead of traditional electrodes [17.28], in which, the liquid metal consumed in the process can be compensated over time, and the capillary containing the liquid metal does not participate in the discharge, thus its shape remains unchanged and solves the problem of electrode wear. Nevertheless, these references reported the experimental characterizations and demonstration of the process based on the preliminary study, and lacked the analysis of liquid metal electrode morphology which will affect the machining accuracy.”

Please see the revised version. Thank you very much.

  • The liquid-metal used in the study is nominated only in raw 281 in the experimental setup and procedure section. I can suppose that the same material has been adopted for the simulation of section 2.2. In this case, if the material properties have been implemented in the numerical model, they should be reported in the section.

Thank you very much for your suggestions. I agree with your comments very much. The same liquid metal electrode material, Galinstan, has been adopted for the simulation and experiment. Based on your suggestions, we have revised and rewritten the paper thoroughly to make the paper more logicality and legibility. Please see the revised version. Thank you very much.

  • The author should also discuss they choice of the liquid metal.

Thank you very much for your suggestions. I have added the reason for choosing Galinstan and made it clearer.

“At present, most of the metals or alloys are in the solid state at room temperature. Exceptions include francium, caesium, rubidium, mercury, sodium–potassium alloys and gallium based alloys, which can be defined as liquid metals. Their melting points are either lower than or close to room temperature, which enable them to remain in the liquid state at room temperature. Unfortunately, the intrinsic radioactivity of caesium, extreme instability of francium rubidium, flammability and corrosivity of sodium–potassium alloy and toxicity of mercury limit their applications to certain specific areas. On the other hand , Gallium based alloy, such as Galinstan (68.5-wt% gallium, 21.5-wt% indium, and 10-wt% tin), a commercially available eutectic liquid alloy, is a low activity and nontoxic liquid with a low melting point (- 19 °C) and low viscosity (0.0024 Pa⋅s at 20 °C) that allows it to be easily transferred through microscale nozzles. Its high electrical conductivity (∼3.5×106 S/m at 20 °C) , high thermal conductivity (16.5 w⋅M-1⋅k-1) and high boiling point (> 1300 °C) are desirable features for a μEDM electrode application. Therefore, Galinstan is used as the liquid electrode in this study”

Please see the revised version. Thank you.

  • In 227- 229, the manuscript reports: “When the extra pressureP1 is 0.1 atm, the liquid metal cannot be extruded from the needle because the extra pressure is too small to overcome the frictional resistance of the capillary inner wall”. How the frictional resistance is modelled in the model? Is it referred to the “no slip wall” condition? If yes, is this condition confirmed by experiments?

I am so sorry about that confusing description.

To simplify the calculation in this manuscript, we chose the no slip wall condition for simulation. (The no-slip boundary condition assumes that the speed of the fluid layer in direct contact with the boundary is identical to the velocity of this boundary. There is no relative movement between the boundary and this fluid layer, therefore there is no slip.) The simulation results show that the liquid metal cannot be extruded from the needle when the extra pressure P1 is 0.1 atm. We also verify this result through experiments. The reason may be that the extra pressure is too small to overcome the internal frictional resistance causing by Galinstan itself viscosity. We did not build the internal frictional resistance model, which is truly difficult to carry out the detailed mathematical analysis and derivation. But it can be confirmed by experiments and didn't affect our analysis.

Based on your suggestions, we have revised the paper thoroughly to make it clearer. Please see the revised version. Thank you very much.

  • Please, check the figure 8, where probably there is a picture over the Fig8(f)

I am so sorry about that confusing description. I have try to revise it and make it clearer.

Please see the revised version. Thank you.

  • About the process.

Is the dielectric fluid in a static condition? In case the kerosene flows between electrode and tool, have the authors found some influence on tool shape?

Thank you very much for your interest in our work.

In the preliminary testing, we found that the flow direction of kerosene would significantly affect machining performance. When patterning a micro groove, if the flushing direction is perpendicular to the direction of pattern, the other side of micro groove will be more flat, and the width of micro groove will become wider than that without flushing. Thus, it shows the kerosene flow will affect the tool shape.

In this manuscript, the dielectric fluid was in a static condition during the Galinstan μEDM process, and another liquid injector is used to periodically flow the EDM fluid to the machining region to flush debris and avoid short circuiting between the liquid electrode and the workpiece.

  • It would be of great interest to evaluate the surface quality of the machined workpiece.

Thank you very much for your interest in our work.

In our future research, we will evaluate the surface quality of the machined workpiece.

Thank you very much for your suggestions.

  • Since the needle must not contribute to the removal process, I suppose that there is a limitation in machining depth that cannot exceed the “h” parameter of fig 2. Moreover, as clearly exposed in the manuscript, the tool electrode shape is assimilable to a cone. This shape does not allow to produce straight wall for the machined geometry. Therefore, even if the presented approach solves the tool wear problem, it seems that there are strong limitations for its adoption. Can the authors list some real case in which the method has been applied?

Thank you very much for your interest in our work.

Under the current conditions, the machining depth seems to depend on the length of the liquid electrode protruding from the needle, and it cannot produce the straight wall geometry. However, if the thickness of the capillary tube wall can be less than the discharge gap, the processing depth will not be limited by the length of the protruding liquid electrode, and the straight wall parts can also be processed (although there are small rounded corners at the bottom).

Fig. I Square microcavity images produced on a silicon substrate using the Galinstan μEDM process(Please see the attachment)

Due to the processing feature of electrode no-wear, it is very suitable for layer-by-layer scanning milling. Square cavity samples (Fig.I) were well prepared using the liquid alloy microelectrode without loss of the electrode (needle) length, which was suggested to potentially overcome the issues related to electrode wear. (The square cavity has a draft angle of several degrees because the liquid electrode is a cone-shaped tip instead of a square tip.) The obtained results encourage further development and optimization of the method toward improved process control and resolution.

Therefore, this work is valuable, and with your support, our research will be better. Thank you.

Thank you for your consideration. I look forward to hearing from you.

Sincerely,

Ruining Huang

School of Mechanical Engineering and Automation

Harbin Institute of Technology, Shenzhen

Shenzhen, China

hrn@hit.edu.cn

Reviewer 2 Report

Dear Authors, 

Many thanks for your interesting contribution. 

I have few request to highlight some aspects of your research:

According to line 85 "The liquid metal consumed in the process can be compensated over time". Would you provide results about consumption of the liquid metal in time when performing empirical experiments?

According to line 124, textual: (the liquid metal does not wet the coated needle). Would you provide a picture of that when you apply pressure (in your experimental research), just before θ=π? (in the supporting information).

In line 170, "Suppose that the liquid metal is the ideal conductor", would you provide some references about ideal conductor regarding your research?

In table 1, what is U?

Author Response

October 8, 2020

Dear Reviewer:

Thank you very much for giving us an opportunity to revise our manuscript. We very much appreciate your efforts to provide constructive comments and suggestions on our manuscript entitled “Investigation of a liquid-phase electrode for micro-electro-discharge machining”.

We have studied your comments carefully. According to your detailed suggestions, we have made a careful revision on the original manuscript. We would like to submit the revised manuscript for your kind consideration. Thank you very much.

The response to your comments are listed as follows.

  • Many thanks for your interesting contribution. 

Thank you very much for your interest in our work.

  • I have few request to highlight some aspects of your research:

According to line 85 "The liquid metal consumed in the process can be compensated over time". Would you provide results about consumption of the liquid metal in time when performing empirical experiments?

Thank you very much for your interest in our work.

Liquid metal feeding control is one critical technique of this method for continuous processing. If the feed volume is less than the consumption volume, then the end of the capillary cannot form a discharge electrode in time, which will lead to a low discharge efficiency. If the feed volume is greater than the consumption volume, then excessive liquid alloy at the end of the capillary will form large droplets that will drip down, thus easily causing short circuits, affecting the process stability, and causing waste. Moreover, the pressure is the most important influencing factor of feed volume in the actual operation. The flow volume (liquid metal feeding volume) will directly impact on material removal rate (MRR) and liquid-metal consumption rate (LMCR). If the relationship between pressure and performance index (MRR and LMCR) is determined, then the relationship between pressure and flow volume is clear. Consequently, the flow volume and pressure comprise a corresponding relationship. The relationship between pressure and MRR and LACR can be directly analyzed. Therefore, the relation of pressure and flow volume can be easy to obtain, and more detailed analysis was carried out in this work.

Thank you very much.

  • According to line 124, textual: (the liquid metal does not wet the coated needle). Would you provide a picture of that when you apply pressure (in your experimental research), just before θ=π? (in the supporting information).

Thank you very much for your interest in our work.

When the contact angle θ is equal or close to π, it means that the liquid metal will eject as shown in fig.7 (e) or Fig.8 (f). A picture (π/2<θ<π) is shown in Fig.II(a). From the picture it can be seen that the liquid metal does not wet the coated needle. It is also found that the coated Parylene C film can eliminate the impact of the discharge on the nozzle tip. On the other hand, the Fig.II(b) shows that liquid metal will wet the stainless steel needle.              

   (a) Coated needle            (b) Without coated needle

Fig.II Pictures of liquid metal on the needle tip: (a) Coated needle; (b) Without coated needle (Please see the attachment)

Thank you very much.

  • In line 170, "Suppose that the liquid metal is the ideal conductor", would you provide some references about ideal conductor regarding your research?

I am so sorry about that confusing description.

The ideal conductor has "ideal" properties, its conductivity is assumed to be infinite (as resistivity approaches 0), which makes calculations easy to perform.

Based on your suggestions, we have revised and rewritten the paper thoroughly to make the paper more legibility. Please see the revised version. Thank you very much.

  • In table 1, what is U?

I am sorry I did not explain it clear in the last version.

The U is the electrostatic field on the boundary when a voltage is applied between liquid metal electrode and workpiece.

I have used the symbol E instead of U to make it clearer. Please see the revised version. Thank you.

Thank you for your consideration. I look forward to hearing from you.

Sincerely,

Ruining Huang

School of Mechanical Engineering and Automation

Harbin Institute of Technology, Shenzhen

Shenzhen, China

hrn@hit.edu.cn

Reviewer 3 Report

The current study deals with a very interesting topic, namely the use of consumable liquid metal electrode for μEDM. Using this kind of electrode a constant tip geometry is ensured, resulting in high dimensional accuracy and repeatability. The tip geometry strongly depends on the applied pressure and voltage. The current study includes a theoretical analysis, along with modeling and simulation of the electrode tip shape for different applied pressures and voltages. Additionally, experiments have been conducted and conclusion were deduced concerning the material removal rate, the electrode consumption rate, and the dimensions of the produced microgrooves.

The paper is well written with robust theoretical background and analysis, thus, it can be considered for publication after some revisions.

More specific:

  • In lines 118-123 a misconception regarding the pressures exists since Pi and Pf are used inconsistently: "In other words, the pressure just outside the surface Pi and just inside the surface Po is equal. Therefore, there is no additional pressure on the plane of the static liquid surface. However, if the pressure inside the surface Pi is greater than the pressure outside the surface Po, ..."
  • In sake of complicity and since no nomenclature is included, all the symbols and constants must be adequately explained. For example in eq. 9 there is not a reference about ε.
  • Figure 4 is very useful and explanatory, nevertheless, the dotted area render it difficult to read.

The simulation is very interesting with impressive results, nevertheless some more analysis is needed. Details concerning the modeling and the thermo-physical properties of the materials have to be mentioned, especially for the alloys. Moreover, the material properties are temperature dependent and in EDM the temperature of the electrode gradually rises. Was this parameter taken into consideration during the simulation? Please provide some more details.

Author Response

October 8, 2020

Dear Reviewer:

Thank you very much for giving us an opportunity to revise our manuscript. We very much appreciate your efforts to provide constructive comments and suggestions on our manuscript entitled “Investigation of a liquid-phase electrode for micro-electro-discharge machining”.

We have studied your comments carefully. According to your detailed suggestions, we have made a careful revision on the original manuscript. We would like to submit the revised manuscript for your kind consideration. Thank you very much.

The response to your comments are listed as follows.

  • The current study deals with a very interesting topic, namely the use of consumable liquid metal electrode for μEDM. Using this kind of electrode a constant tip geometry is ensured, resulting in high dimensional accuracy and repeatability. The tip geometry strongly depends on the applied pressure and voltage. The current study includes a theoretical analysis, along with modeling and simulation of the electrode tip shape for different applied pressures and voltages. Additionally, experiments have been conducted and conclusion were deduced concerning the material removal rate, the electrode consumption rate, and the dimensions of the produced microgrooves.

The paper is well written with robust theoretical background and analysis, thus, it can be considered for publication after some revisions.

Thank you very much for your interest in our work.

  • More specific:

In lines 118-123 a misconception regarding the pressures exists since Pi and Pf are used inconsistently: "In other words, the pressure just outside the surface Pi and just inside the surface Po is equal. Therefore, there is no additional pressure on the plane of the static liquid surface. However, if the pressure inside the surface Pi is greater than the pressure outside the surface Po, ..."

I am so sorry about that confusing description. Thank you very much for your suggestions. I have revised it and made it clearer as follows.

“In other words, the pressure just outside the surface Po and just inside the surface Pi is equal. Therefore, there is no additional pressure on the plane of the static liquid surface. However, if the pressure inside the surface Pi is greater than the pressure outside the surface Po, an additional pressure Pf will be generated and induce a curved liquid surface…”

Please see the revised version. Thank you very much.

  • In sake of complicity and since no nomenclature is included, all the symbols and constants must be adequately explained. For example in eq. 9 there is not a reference about ε.

Thank you very much for your suggestions. I have added explanation for the symbols ε.

“…, and ε is dielectric constant.”

Please see the revised version. Thank you very much.

  • Figure 4 is very useful and explanatory, nevertheless, the dotted area render it difficult to read.

Thank you very much for your suggestions. I have changed the color and transparency of the dot and made it clearer.

Please see the revised version. Thank you very much.

  • The simulation is very interesting with impressive results, nevertheless some more analysis is needed. Details concerning the modeling and the thermo-physical properties of the materials have to be mentioned, especially for the alloys. Moreover, the material properties are temperature dependent and in EDM the temperature of the electrode gradually rises. Was this parameter taken into consideration during the simulation? Please provide some more details.

Thank you very much for your suggestions.

We have added the properties of the liquid metal for simulation in the manuscript. Please see the revised version.

In the simulation, we did not consider the EDM discharge and temperature, and specifically focused on the formation process of liquid metal electrode before discharge. Therefore, the phase-field modeling was adopted, and only the two most important factors affecting the morphology of liquid metal electrodes, pressure and electric field, were discussed.

As you suggested, The EDM process will generate heat and affect the morphology of the electrodes. However, it is truly difficult to model and simulate the whole EDM process if all the factors are considered. In our future research, we will try to simulate the whole EDM process.

Thank you very much for your suggestions, and with your support, our research will be better.

Thank you for your consideration. I look forward to hearing from you.

Sincerely,

Ruining Huang

School of Mechanical Engineering and Automation

Harbin Institute of Technology, Shenzhen

Shenzhen, China

hrn@hit.edu.cn
